# Return Augmented Decision Transformer for Off-Dynamics Reinforcement Learning

## Abstract

We study offline off-dynamics reinforcement learning (RL) to utilize data from an easily accessible source domain to enhance policy learning in a target domain with limited data. Our approach centers on return-conditioned supervised learning (RCSL), particularly focusing on Decision Transformer (DT) type frameworks, which can predict actions conditioned on desired return guidance and complete trajectory history. Previous works address the dynamics shift problem by augmenting the reward in the trajectory from the source domain to match the optimal trajectory in the target domain. However, this strategy can not be directly applicable in RCSL owing to (1) the unique form of the RCSL policy class, which explicitly depends on the return, and (2) the absence of a straightforward representation of the optimal trajectory distribution. We propose the Return Augmented (REAG) method for DT type frameworks, where we augment the return in the source domain by aligning its distribution with that in the target domain. We provide the theoretical analysis demonstrating that the RCSL policy learned from REAG achieves the same level of suboptimality as would be obtained without a dynamics shift. We introduce two practical implementations $\text{REAG}^*_{\text{Dara}}$ and $\text{REAG}^*_{\text{MV}}$ respectively. Thorough experiments on D4RL datasets and various DT-type baselines demonstrate that our methods consistently enhance the performance of DT type frameworks in off-dynamics RL.

## 1 Introduction

Off-dynamics reinforcement learning (Eysenbach et al., 2020; Jiang et al., 2021; Liu et al., 2022; Liu and Xu, 2024; Guo et al., 2025) arises in decision-making domains such as autonomous driving (Pan et al., 2017) and medical treatment (Laber et al., 2018; Liu et al., 2023), where direct policy training through trial-and-error in the target environment is often costly, unethical, or infeasible. A common strategy is to train the policy in source environments with similar but more accessible dynamics. However, discrepancies between the source and target environments create a simulation-to-reality (sim-to-real) gap, which can lead to catastrophic failures when deploying the source-trained policy in the target environment.

Beyond the challenge of dynamics shift, practical scenarios often do not allow real-time online interaction with the source environment due to time and computational constraints. As a result, policies must be learned from pre-collected datasets generated by behavior policies. This setting is particularly difficult, as it combines off-policy, offline, and off-dynamics characteristics. Recently, supervised learning–based methods (Chen et al., 2021; Brandfonbrener et al., 2022) have emerged as more stable and scalable alternatives to traditional offline reinforcement learning algorithms grounded in dynamic programming (Levine et al., 2020). In the offline off-dynamics setting, the majority of training data is drawn from the source domain, with only a limited portion collected from the target

domain. Our study focuses on advancing Decision Transformer (DT) type frameworks (Chen et al., 2021; Hu et al., 2024; Zhuang et al., 2024) for off-dynamics reinforcement learning, which can be viewed as a special case of return-conditioned supervised learning (RCSL) (Emmons et al., 2021; Brandfonbrener et al., 2022). While DT-type methods have gained significant attention across various reinforcement learning tasks, no prior work has explicitly tackled the off-dynamics RL problem.

There are several previous significant works in off-dynamics reinforcement learning that employ reward augmentation to address the dynamics shift between source and target environments (Eysenbach et al., 2020; Liu et al., 2022). In particular, Eysenbach et al. (2020) proposed the DARC algorithm to train a policy in the source domain using augmented rewards. These augmentations are derived by minimizing the KL distance between the distribution of trajectories generated by the learning policy in the source domain and those generated by the optimal policy in the target domain. Liu et al. (2022) extended this idea to the offline setting with the DARA algorithm. However, these reward augmentation techniques for dynamic programming based RL algorithms are not directly applicable to RCSL methods for two primary reasons. First, the policy classes used in RCSL methods explicitly depend on the conditional return-to-go function, leading to different trajectory distributions that invalidate the trajectory matching methods. Second, the augmentation techniques in Eysenbach et al. (2020); Liu et al. (2022) explicitly rely on the form of the optimal trajectory distribution in the target domain. In contrast, there is no straightforward representation of the optimal RCSL policy and the trajectory distribution. Therefore, novel augmentation mechanisms must be derived for RCSL methods to effectively address off-dynamics reinforcement learning.

In this work, we propose the Return Augmented (REAG) algorithm, which augments the returns of trajectories from the source environment to align with the target environment in DT type framework. Through rigorous analysis, we show that the RCSL policy learned with REAG in the source domain achieves suboptimality comparable to that learned directly in the target domain without dynamics shift. Specifically, our contributions are summarized as follows:

- We propose a novel method, Return Augmented (REAG), designed specifically for DT-type algorithms. The approach augments the returns of offline trajectories in the source domain by leveraging a small amount of data from the target domain. We develop two practical implementations of REAG: $\text{REAG}^*_{\text{Dara}}$, derived from reward augmentation techniques used in dynamic programming–based methods, and $\text{REAG}^*_{\text{MV}}$ from direct return distribution matching.
- We provide a rigorous theoretical analysis demonstrating that the return-conditioned policy learned from REAG can achieve the same suboptimality as a policy learned directly from the target domain. Our analysis relies on the same data coverage assumptions made by Brandfonbrener et al. (2022) where there is no dynamics shift, implying that return augmentation could enhance the performance of RCSL in off-dynamics RL when the available source dataset size is much larger than the available target dataset size.
- We conduct experiments on the D4RL benchmark by training policies on source datasets collected from modified dynamics and evaluating them in the original environments. Across DT-type baselines—including DT (Chen et al., 2021), Reinformer (Zhuang et al., 2024) and QT (Hu et al., 2024)—both $\text{REAG}^*_{\text{Dara}}$ and $\text{REAG}^*_{\text{MV}}$ consistently improve performance, with $\text{REAG}^*_{\text{MV}}$ showing the greatest gains, highlighting the advantage of return-level augmentation.

## 2 Preliminary

**Sequential Decision-Making.** We consider a general sequential decision-making problem. At each step $t$, the agent receives an observation $o_t$ from the environment. Based on the history up to step $t$, the agent makes action $a_t$ and receives the reward $r_t$ from the environment. The agent interacts with the environment in episodes with a length $H$. We use $\tau = (o_1, a_1, r_1, \cdots, o_H, a_H, r_H)$ to denote a whole trajectory, and we use $g(\tau) = \sum_{t=1}^H r_t$ to denote the cumulative return of the trajectory. We model the environment as a Markov Decision Process (MDP) $M$, which consists of $(\mathcal{S}, \mathcal{A}, p, r, H)$. Here $\mathcal{S}$ is the state space, each state $s$ represents the possible history up to some time step $t$, i.e., $s = (o_1, a_1, r_1, \cdots, o_t)$. $\mathcal{A}$ is the action space, $p(s'|s, a)$ is the transition dynamics that determines the transition probability for the agent to visit state $s'$ from current state $s$ with the action $a$. $r(s, a)$ denotes the reward function. We re-define a trajectory as $\tau = (s_1, a_1, r_1, \cdots, s_H, a_H, r_H)$. We assume that each $s$ corresponds to one single time step $t = t(s)$, and we denote $g_\pi(s) = \mathbb{E}_{\tau \sim \pi}[g(\tau)|s_1 = s]$. Then the goal of the agent is to learn a policy $\pi : \mathcal{S} \to \mathcal{A}$ that maximizes the expected accumulated reward $J(\pi) := \mathbb{E}_{\tau \sim \pi}[g(\tau)]$. We denote the optimal policy as $\pi^*$.

**Offline RL and Decision Transformer.** We consider the offline reinforcement learning setting. Given a dataset $\mathcal{D}$, the goal of the agent is to learn $\pi^*$ from $\mathcal{D}$. We assume that the trajectories in $\mathcal{D}$ are generated from a behavior policy $\beta$. In this work, we mainly consider Decision Transformer (DT) (Chen et al., 2021) as our backbone algorithm. DT is a type of sequential modeling technique based on Transformer (Vaswani et al., 2017) to solve offline RL problems. In detail, DT maintains a function $\pi(a|s, g)$ as its policy function. To train the plolicy $\pi$, DT aims to minimize the following negative log-likelihood function $\hat{L}(\pi) := \hat{L}(\pi) := -\sum_{\tau \in \mathcal{D}} \sum_{1 \le t \le H} \log \pi(a_t|s_t, g(\tau))$. To evaluate $\pi$, DT defines a *conditioning function* $f : \mathcal{S} \to \mathbb{R}$, which maps each state to a return value and guides the policy $\pi_f$ within the environment, where $\pi_f(a|s) := \pi(a|s, f(s))$. The conditioning function is pivotal in DT, as varying $f(s)$ for a given state $s$ results in different policies. To achieve the optimal policy, $f(s)$ should be maximized (Zhuang et al., 2024).

**Offline Off-Dynamics RL.** In this work, we consider the offline off-dynamics RL problem, where the agent has access to two offline datasets $\mathcal{D}^S$ and $\mathcal{D}^T$. $\mathcal{D}^S, \mathcal{D}^T$ include the data collected from the *source environment* $M^S$ and the *target environment* $M^T$. The source and the target environments share the same reward function $r$, with different transition dynamics $p^S$ and $p^T$. In practice, we assume that the dataset size from the source dataset $|\mathcal{D}^S|$ is much larger than the data coming from the target dataset $|\mathcal{D}^T|$. Then the agent aims to find the optimal policy for the target environment $M^T$ based on the data from both the source and the target environments. Since the transition dynamics $p^S$ and $p^T$ are different, we can not directly apply existing RL algorithms on the union $\mathcal{D}^S \cup \mathcal{D}^T$.

# 3 Return Augmentation for Goal Conditioned Supervised Learning

## 3.1 Return-Augmented Framework

DT has the potential to address offline off-dynamics reinforcement learning challenges, as shown in Table 1. However, it still has certain limitations. To overcome these, we propose a general framework that efficiently learns the optimal policy for the target environment using the combined dataset $\mathcal{D}^S \cup \mathcal{D}^T$. Leveraging the return-conditioning nature of DT, we introduce a *return augmentation* technique that modifies returns in the offline source dataset through a transformation function. This approach allows the policy derived from the augmented source dataset to effectively approximate the optimal policy of the target environment, as illustrated in the following equation, where $\pi^S$ represents a strong candidate for approximating the optimal policy of the target environment and $\psi$ is the carefully chosen transformation function.

$$\pi^S = \arg\min_\pi \hat{L}(\pi) := -\sum_{\tau \in \mathcal{D}^S} \sum_{1 \le t \le H} \log \pi(a_t|s_t, \psi(g(\tau))).$$

We call our method Return Augmented (REAG) for DT. Next we introduce two methods to construct $\psi$, based on the dynamics-aware reward augmentation (DARA) technique (Eysenbach et al., 2020; Liu et al., 2022), and a direct return distribution matching method.

## 3.2 Dynamics-Aware Reward Augmentation

We first summarize the idea of DARA. Let $p^T(s'|s, a)$ denote the transition dynamics of the target environment, and $p^S(s'|s, a)$ denote the source environment. According to the connection of RL and probabilistic inference (Levine, 2018), we can turn the optimal policy finding problem into an inference problem. We use $O$ to denote a binary random variable where $O = 1$ suggests $\tau$ is a trajectory induced by the optimal policy. Given a trajectory $\tau$, the likelihood of $\tau$ being a trajectory induced by the optimal policy under the target environment is $p^T(O = 1|\tau) = \exp(\sum_{t=1}^H r(s_t, a_t)/\eta)$, where $\eta$ is the step size parameter used for tuning. It means that the trajectory with higher cumulative rewards is more likely to be the trajectory induced by the optimal policy. We introduce a variational distribution $p_\pi^S(\tau) = p(s_1) \prod_{t=1}^T p^S(s_{t+1}|s_t, a_t)\pi(a_t|s_t)$ to approximate $p_\pi^T(O = 1|\tau)$. Then we have

$$\log p_\pi^T(O = 1) = \log \mathbb{E}_{\tau \sim p_\pi^T(\tau)} p^T(O = 1|\tau)$$
$$\ge \mathbb{E}_{\tau \sim p_\pi^S(\tau)} \big[ \log p^T(O = 1|\tau) + \log \big(p_\pi^T(\tau)/p_\pi^S(\tau)\big)\big]$$
$$= \mathbb{E}_{\tau \sim p_\pi^S(\tau)} \big[ \sum_{t=1}^T r(s_t, a_t)/\eta - \log \big(p^S(s_{t+1}|s_t, a_t)/p^T(s_{t+1}|s_t, a_t)\big)\big], \quad (3.1)$$

where for the first inequality, we change the distribution of the expectation from $P_\pi^T(\tau)$ to $P_\pi^S(\tau)$ and then use Jensen's inequality to derive the result; the second equation holds due to the assumption/-modeling that the likelihood of $\tau$ being a trajectory induced by the optimal policy under the target

environment is $P^T(O = 1|\tau) = \exp(\sum_{t=1}^{H} r(s_t, a_t)/\eta)$. Therefore, we obtain an evidence lower bound of $p_\pi^T(O = 1)$, which equals to find a policy to maximize the value in the source environment, with the augmented reward $r^S(s_t, a_t) = r(s_t, a_t) + \eta \log p^T(s_{t+1}|s_t, a_t) - \eta \log p^S(s_{t+1}|s_t, a_t)$. Following Eysenbach et al. (2020), to estimate the $\log p^T(s_{t+1}|s_t, a_t) - \log p^S(s_{t+1}|s_t, a_t)$, we use a pair of learned binary classifiers which infers whether the transitions come from the source or target environments. Specifically, we denote classifiers $q_{sas}(\cdot|s, a, s')$ and $q_{sa}(\cdot|s, a)$, which return the probability for some $(s, a, s')$ or $(s, a)$ tuples whether they belong to the source or the target environments. Then according to Eysenbach et al. (2020), we have

$$\log p^T(s_{t+1}|s_t, a_t) - \log p^S(s_{t+1}|s_t, a_t) = \Delta r(s_t, a_t, s_{t+1})$$
$$:= \log \frac{q(M^T|s_t, a_t, s_{t+1})}{q(M^S|s_t, a_t, s_{t+1})} - \log \frac{q_{sa}(M^T|s_t, a_t)}{q_{sa}(M^S|s_t, a_t)}. \quad (3.2)$$

For a trajectory $\tau = (s_1, a_1, r_1, ..., s_H, a_H, r_H)$, we denote the transformation $\psi(g(s_t)) := \sum_{h=t}^{H} r_h + \eta \sum_{h=t}^{H} \Delta r(s_h, a_h, s_{h+1})$. We denote such a transformation method as $\text{REAG}_{\text{Dara}}^*$.

### 3.3 Direct Matching of Return Distributions

The reward augmentation strategy in $\text{REAG}_{\text{Dara}}^*$ stems from the probabilistic inference view of RL which matches the distribution of the learning trajectory in the source domain with that of the optimal trajectory in the target domain (Eysenbach et al., 2020). However, it does not fully capture the power of DT, which is able to induce a *family of policies* that are conditioned on the return-to-go $f$. By varying $f$, DT enables the generation of a diverse range of policies, including the optimal one. In contrast, $\text{REAG}_{\text{Dara}}^*$ assumes a single, fixed target policy, and thus its augmentation strategy cannot generalize across multiple policies induced by varying $f$ in DT. As a result, it cannot find the desired return conditioned policy when evaluated with a different $f$ in the target domain. This motivates us to find a return transformation method $\psi$ to guarantee that $\pi_f^S(a|s) \approx \pi_f^T(a|s)$ for all $f$.

We consider a simplified case where both $D^S$ and $D^T$ are generated by following the same behavior policy $\beta(a|s)$. We use $d_S(A)$ and $d_T(A)$ to denote the probability for event $A$ to happen under the source and target environments following $\beta$. With a slight abuse of notation, we use $g_S$ and $g_T$ to denote the return following the behavior policy. Then we characterize the learned policies by DT under the infinite data regime (Brandfonbrener et al., 2022) for both the source environment and target environment. According to Brandfonbrener et al. (2022), $\pi_f^S(a|s) = P^S(a|s, \psi(g_S) = f(s))$. Then we can express $\pi^S$ and $\pi^T$ as

$$\pi_f^S(a|s) = \frac{d_S(a|s)d_S(\psi(g_S) = f(s)|s, a)}{d_S(\psi(g_S) = f(s)|s)}, \quad \pi_f^T(a|s) = \frac{d_T(a|s)d_T(g_T = f(s)|s, a)}{d_T(g_T = f(s)|s)}.$$

Since the behavior policies over the source and target environments are the same, we have $d_S(a|s) = d_T(a|s)$ for all $(s, a)$. Then in order to guarantee $\pi_f^S(a|s) = \pi_f^T(a|s)$ we only need to guarantee $d_S(\psi(g_S(s)) = \cdot|s, a) = d_T(g_T(s) = \cdot|s, a)$, $\forall s, a$. Denote the cumulative distribution function (CDF) of $g^S$ conditioned on $s, a$ is $g^S|s, a \sim G_\beta^S(s, a)$, and $g^T|s, a \sim G_\beta^T(s, a)$. Then if both $G_\beta^S(s, a)$ and $G_\beta^T(s, a)$ are invertible, we can set $\psi$ as follows

$$\psi(g^S) = G_\beta^{T, -1}(G_\beta^S(g_S; s, a); s, a). \quad (3.3)$$

If there exist $P^S$, $P^T$, and $r$ such that the DARA-type augmented reward-to-go satisfies (3.3), then the DARA-type reward augmentation can be deemed as a special case of the transformation (3.3). In general, $G_\beta^T$ and $G_\beta^S$ are hard to obtain and computationally intractable, making $\psi$ intractable either. We use Laplace approximation to approximate both $G_\beta^T$ and $G_\beta^S$ by Gaussian distributions, e.g., $G_\beta^S(s, a) \sim N(\mu^S(s, a), \sigma_S^2(s, a))$ and $G_\beta^T(s, a) \sim N(\mu^T(s, a), \sigma_T^2(s, a))$. We then obtain that

$$\psi(g^S) := \frac{g^S - \mu^S(s, a)}{\sigma^S(s, a)} \cdot \sigma^T(s, a) + \mu^T(s, a). \quad (3.4)$$

We denote DT with a $\psi$ transformation from (3.4) by $\text{REAG}_{\text{MV}}^*$, since such a transformation only depends on the estimation of mean values $\mu^S, \mu^T$ and variance $\sigma^S, \sigma^T$.

## 3.4 Sample Complexity of Off-Dynamics RCSL

In this section, we provide an overview of the sample complexity for off-dynamics RCSL. Let $N^S$ represent the number of trajectories in the source dataset $\mathcal{D}^S$ and $N^T$ the number of trajectories in the target dataset $\mathcal{D}^T$. We define $J^T(\pi)$ as the expected cumulative reward under any policy $\pi$ within the target environment. Our theorem is established based on the following assumptions.

**Assumption 3.1.** (1) (Return coverage) $P_\beta^T(g = f(s_1)|s_1) \geq \alpha_f$ for all initial states $s_1$. (2) (Near determinism) $P(r \neq r(s,a)$ or $s' \neq T(s,a)|s,a) \leq \epsilon$ at all $s, a$ for some functions $T$ and $r$. (3) (Consistency of $f$) $f(s) = f(s') + r$ for all $s$.

**Assumption 3.2.** For all $s$ we assume (1) (Bounded occupancy mismatch) $P_{\pi_f^{\text{RCSL}}(s)} \leq C_f P_\beta(s)$; (2) (Return coverage) $P_\beta^T(g = f(s)|s) \geq \alpha_f$; and (3) (Domain occupancy overlap) $d_\beta^T(s) \leq \gamma_f d_\beta^S(s)$.

**Assumption 3.3.** (1) The policy class $\Pi$ is finite. (2) $|\log \pi(a|s,g) - \log \pi(a'|s',g')| \leq c$ for any $(a, s, g, a', s', g')$ and all $\pi \in \Pi$. (3) The approximation error is bounded by $\epsilon_{\text{approx}}$, i.e., $\min_{\pi \in \Pi} L(\pi) \leq \epsilon_{\text{approx}}$.

Assumptions 3.1 to 3.3 are the same as the assumptions imposed in Theorem 1, Theorem 2, and Corollary 3 in Brandfonbrener et al. (2022) respectively. Now we present our theoretical result.

**Theorem 3.4.** Under Assumptions 3.1 to 3.3 on the coverage of the offline dataset and the occupancy overlap of the source and target environments, with high probability, we have $J^T(\pi^\star) - J^T(\hat{\pi}_f) = O\big(1/(N^T + N^S)^{1/4}\big)$, where $O$ omits terms that are independent of the sample size $N^T$ of the target domain and the sample size $N^S$ of the source domain.

**Remark 3.5.** Theorem 3.4 suggests that the modified samples from the source domain could enhance the performance of RCSL, for which the sample complexity is approximately $O((1/N^T)^{1/4})$.

For more theoretical details, please refer to Appendix B.

# 4 Experiments

In this section, we first outline the fundamental setup of the experiment. We then describe experiments designed to address specific questions, with each question and its corresponding answer detailed in a separate subsection.

- How effective are DT-type methods in mitigating the impact of limited data in target environment?
- What techniques can be employed to improve the performance of DT-type methods in off-dynamics scenarios while addressing the constraints of offline data shortages in target environment?
- How does the performance of DT-type methods compare to baselines in off-dynamics problems?

## 4.1 Basic Experiment Setting

**Tasks and Environments.** We study established D4RL tasks in the Gym-MuJoCo environment (Fu et al., 2020), a suite built atop the MuJoCo physics simulator, featuring tasks such as locomotion and manipulation. Particularly, we focused on three environments: Walker2D, Hopper, and HalfCheetah. In addressing the off-dynamics reinforcement learning problem, we distinguish between the Source and Target environments. The Target environment is based on the original Gym-MuJoCo framework, while the Source environment is modified using two shift methods: BodyMass Shift and JointNoise Shift. In the BodyMass Shift, the mass of the body is altered in the Source environment, whereas in the JointNoise Shift, random noise is added to the actions.

**Dataset.** For the Target Dataset corresponding to the Target Environment, we leverage the official D4RL data to construct the target datasets: 10T and 1T. The 10T dataset comprises ten times the number of trajectories compared to the 1T dataset.[1] For the Source Dataset collection, we begin by

---

[1]Unlike the approach of Liu et al. (2022), which constructs the 1T dataset by selecting the last 1/10 timesteps from the original target dataset (10T), we propose a uniform sampling method across trajectories in the target dataset.

| | BEAR | | | AWR | | | BCQ | | | CQL | | |
|---|---|---|---|---|---|---|---|---|---|---|---|---|
| | **M** | **M-R** | **M-E** | **M** | **M-R** | **M-E** | **M** | **M-R** | **M-E** | **M** | **M-R** | **M-E** |
| **1T** | $4.638_{\pm3.882}$ | $0.777_{\pm0.105}$ | $9.267_{\pm1.692}$ | $68.023_{\pm1.687}$ | $28.426_{\pm2.974}$ | $100.566_{\pm0.513}$ | $62.567_{\pm2.459}$ | $60.638_{\pm0.683}$ | $101.610_{\pm1.309}$ | $65.618_{\pm2.818}$ | $57.402_{\pm6.161}$ | $101.611_{\pm0.143}$ |
| **10T** | $13.143_{\pm3.016}$ | $5.852_{\pm0.168}$ | $21.383_{\pm1.237}$ | $78.060_{\pm0.772}$ | $58.286_{\pm1.684}$ | $109.154_{\pm0.976}$ | $74.735_{\pm1.184}$ | $64.735_{\pm2.555}$ | $101.840_{\pm1.962}$ | $78.191_{\pm1.839}$ | $80.145_{\pm2.286}$ | $101.840_{\pm0.467}$ |

| | MOPO | | | DT | | | Reinformer | | | QT | | |
|---|---|---|---|---|---|---|---|---|---|---|---|---|
| | **M** | **M-R** | **M-E** | **M** | **M-R** | **M-E** | **M** | **M-R** | **M-E** | **M** | **M-R** | **M-E** |
| **1T** | $20.953_{\pm2.715}$ | $20.313_{\pm3.488}$ | $20.569_{\pm0.983}$ | $67.261_{\pm2.316}$ | $34.482_{\pm5.890}$ | $107.171_{\pm1.611}$ | $79.034_{\pm1.506}$ | $38.072_{\pm9.174}$ | $103.284_{\pm5.437}$ | $81.756_{\pm1.671}$ | $67.546_{\pm9.505}$ | $111.722_{\pm1.398}$ |
| **10T** | $22.261_{\pm2.811}$ | $18.529_{\pm1.760}$ | $21.196_{\pm3.103}$ | $79.697_{\pm3.348}$ | $68.528_{\pm1.924}$ | $108.622_{\pm1.815}$ | $81.377_{\pm1.903}$ | $68.168_{\pm2.661}$ | $109.845_{\pm0.726}$ | $88.262_{\pm12.886}$ | $85.092_{\pm8.727}$ | $111.394_{\pm0.469}$ |

**Table 1** Performance comparison of algorithms on the **1T**, **10T**, and **1T10S** datasets. In this study, **1T10S(B)** refers to the source dataset under the **BodyMass shift** setting, while **1T10S(J)** corresponds to the source dataset under the **JointNoise shift** setting. Experiments are conducted using the **Medium (M)**, **Medium-Replay (M-R)**, and **Medium-Expert (M-E)** datasets. We present the results for the **Walker2D** environment here; complete results are provided in appendix E. All reported values are averaged over five seeds (0, 1012, 2024, 3036, 4048).

modifying the environment through adjustments to the XML file of the MuJoCo simulator. We then collect the Random, Medium, Medium-Replay, and Medium-Expert offline datasets in the modified environments, following the same data collection procedure as used in D4RL. For further details on the dataset collection process and the datasets, please refer to the Appendix D.

**Baselines.** In selecting our baseline models, we incorporate a diverse set of well-established off-dynamics RL methods, including BEAR (Kumar et al., 2019), AWR (Peng et al., 2019), BCQ (Fujimoto et al., 2019), CQL (Kumar et al., 2020), and MOPO (Yu et al., 2020). Furthermore, we enhance these baseline models by incorporating DARA augmentation, resulting in augmented algorithms that also serve as baselines for comparison with our proposed method. In establishing hyperparameters, we ensure consistency across tasks for certain parameters, such as the learning rate and the number of iteration steps. Refer to Appendix D for further details on the parameter settings.

## 4.2 Evaluation of Adaptability and Data Efficiency in RCSL Algorithms

We evaluate three representative DT-type algorithms include DT (Chen et al., 2021), Reinformer (Zhuang et al., 2024) and QT (Hu et al., 2024) to assess their ability to enable an adaptive policy while reducing reliance on offline data in the target environment. To conduct this evaluation, we perform two experiments: (1) We examine the performance of the three DT-type algorithms under varying dataset sizes and quality levels in the target environment; (2) We evaluate their effectiveness in off-dynamics scenarios.

To assess the impact of dataset size and quality on the performance of DT-type algorithms, we evaluate three algorithms using two datasets: a subset of the target data (1T) and the full target dataset (10T), comparing the results against other baselines. These experiments aim to quantify the performance gap between training on 1T and 10T datasets, highlighting the effects of target environment data scarcity and establishing a benchmark for off-dynamics settings. In off-dynamics offline RL, instead of relying solely on a large target dataset, we incorporate a small subset of target data with a larger source dataset. To examine how effectively algorithms leverage source data, we construct the 1T10S dataset by combining a subset of target data (1T) with the full source dataset (10S), following the setting of Liu et al. (2022). This dataset serves as the training set for DT-type algorithms, whose performance is then evaluated in the target environment. For a comprehensive comparison, we benchmark DT-type algorithms against other baseline methods.

The evaluation results in Table 1 demonstrate the impact of dataset size and off-dynamics settings on algorithm performance. With limited training data, the algorithm's learning capacity is restricted, leading to degraded performance, especially when target-environment data are scarce. To mitigate this issue, we incorporate additional source datasets under BodyMass Shift and JointNoise Shift settings, which improve generalization to the target environment. However, while leveraging source data can partially compensate for the shortage of target data, it remains less effective than training with sufficient target-environment data. To further improve DT-type frameworks under off-dynamics settings, we propose two return-based augmentation methods, $REAG^*_{MV}$ and $REAG^*_{Dara}$, which can be applied to DT, Reinformer, and QT frameworks. Specifically, applying $REAG^*_{MV}$ yields $\textbf{REAG}^{\textbf{DT}}_{\textbf{MV}}$, $\textbf{REAG}^{\textbf{Reinf}}_{\textbf{MV}}$, and $\textbf{REAG}^{\textbf{QT}}_{\textbf{MV}}$, while applying $REAG^*_{Dara}$ produces $\textbf{REAG}^{\textbf{DT}}_{\textbf{Dara}}$, $\textbf{REAG}^{\textbf{Reinf}}_{\textbf{Dara}}$, and $\textbf{REAG}^{\textbf{QT}}_{\textbf{Dara}}$, demonstrating the promise of these augmentation techniques in enhancing algorithm performance under off-dynamics conditions.

| | | | DT | | | Reinformer | | | QT | | |
|---|---|---|---|---|---|---|---|---|---|---|---|
| | | | 1T10S | REAG$_{MV}^{DT}$ | REAG$_{Dara}^{DT}$ | 1T10S | REAG$_{MV}^{Reinf}$ | REAG$_{Dara}^{Reinf}$ | 1T10S | REAG$_{MV}^{QT}$ | REAG$_{Dara}^{QT}$ |
| W2D | M | BM | 78.768±1.233 | 80.857±1.715↑ | 78.257±2.423↓ | 80.857±0.509 | 82.354±1.479↑ | 80.666±0.505↓ | 84.325±0.425 | 84.582±0.507↑ | 83.068±0.859↓ |
| | | JN | 71.068±1.022 | 75.008±1.834↑ | 71.779±1.706↑ | 74.748±1.721 | 75.008±0.986↑ | 74.268±1.341↓ | 80.621±1.143 | 80.904±1.502↑ | 78.672±2.201↓ |
| | M-R | BM | 73.664±1.920 | 73.708±1.570↑ | 67.565±0.799↓ | 67.032±5.767 | 50.296±14.211↓ | 66.658±4.303↓ | 87.292±0.631 | 87.491±1.226↑ | 76.169±7.567↓ |
| | | JN | 58.255±3.181 | 55.722±2.653↓ | 62.226±0.383↑ | 54.801±3.217 | 47.591±10.244↓ | 55.438±4.833↑ | 82.139±1.029 | 82.363±4.206↑ | 79.795±4.708↓ |
| | M-E | BM | 84.430±0.823 | 88.235±1.886↑ | 85.328±0.865↑ | 83.388±0.806 | 84.897±1.117↑ | 83.761±0.735↓ | 93.082±0.348 | 92.744±0.499↓ | 94.578±1.383↑ |
| | | JN | 115.746±1.116 | 111.060±2.247↓ | 111.236±0.914↓ | 117.360±2.550 | 118.218±1.460↑ | 117.765±2.499↑ | 116.149±1.640 | 118.564±0.697↑ | 116.115±1.889↓ |
| Hp | M | BM | 34.057±0.177 | 39.435±1.239↑ | 37.787±1.914↑ | 51.357±3.713 | 59.085±2.791↑ | 51.771±5.322↑ | 49.516±9.798 | 51.796±9.971↑ | 62.262±5.348↑ |
| | | JN | 70.685±0.726 | 70.356±3.657↓ | 78.325±2.522↑ | 70.340±4.633 | 72.346±5.877↑ | 70.466±3.728↑ | 68.656±7.079 | 73.987±8.080↑ | 68.709±12.160↑ |
| | M-R | BM | 64.216±1.504 | 66.092±0.233↑ | 60.393±1.086↓ | 17.534±6.725 | 20.952±9.794↑ | 27.238±12.735↑ | 69.460±13.948 | 76.287±7.810↑ | 82.786±11.992↑ |
| | | JN | 61.870±0.249 | 77.825±1.638↑ | 83.525±1.728↑ | 41.820±15.773 | 43.985±5.075↑ | 52.052±10.035↑ | 93.704±7.559 | 93.409±4.696↓ | 51.456±12.168↓ |
| | M-E | BM | 33.554±0.846 | 52.873±0.454↑ | 33.631±1.605↑ | 68.973±7.512 | 64.206±12.073↓ | 73.363±7.674↑ | 61.162±3.767 | 73.952±16.294↑ | 77.279±18.607↑ |
| | | JN | 108.254±1.583 | 109.367±1.084↑ | 108.261±2.612↑ | 109.256±0.126 | 109.472±0.103↑ | 109.255±0.188↓ | 109.056±0.214 | 109.803±0.609↑ | 109.746±0.771↑ |
| Hc | M | BM | 39.954±0.260 | 40.250±0.911↑ | 37.599±0.395↓ | 37.353±0.483 | 42.451±0.491↑ | 38.261±1.238↑ | 44.656±0.643 | 47.303±0.318↑ | 46.383±0.358↑ |
| | | JN | 47.725±0.431 | 44.149±3.672↓ | 47.833±0.284↑ | 48.274±0.191 | 43.009±0.307↓ | 48.404±0.168↑ | 56.213±0.327 | 52.394±1.413↓ | 55.026±0.410↓ |
| | M-R | BM | 20.966±9.607 | 27.812±3.256↑ | 24.059±2.271↑ | 31.584±1.248 | 32.114±1.455↑ | 26.995±4.373 | 41.300±0.787 | 42.405±0.729↑ | 41.359±0.985↑ |
| | | JN | 36.509±4.414 | 38.417±4.068↑ | 38.031±3.529↑ | 40.296±2.914 | 40.840±2.880↑ | 38.436±3.377↓ | 53.763±0.793 | 53.870±0.981↑ | 53.257±0.586↓ |
| | M-E | BM | 54.981±1.147 | 56.228±2.930↑ | 51.357±8.231↓ | 40.568±0.984 | 46.048±1.657↑ | 55.818±1.849↑ | 71.080±8.802 | 69.819±5.120↓ | 76.533±8.022↑ |
| | | JN | 70.573±8.599 | 77.762±2.099↑ | 77.751±2.702↑ | 76.073±3.878 | 79.390±0.149↑ | 78.981±1.198↓ | 82.961±4.019 | 83.692±0.699↑ | 82.148±2.758↓ |

**Table 2** Performance evaluation of two return augmentation methods, **REAG$_{MV}^*$** and **REAG$_{Dara}^*$**, integrated with **DT**, **Reinformer**, and **QT** frameworks in off-dynamics scenarios. The experiments are conducted in the **Walker2D (W2D)**, **Hopper (Hp)**, and **HalfCheetah (Hc)** environments under the **Medium (M)**, **Medium-Replay (M-R)**, and **Medium-Expert (M-E)** settings. The source environment is modified using two shift conditions: **BodyMass shift (BM)** and **JointNoise shift (JN)**. For reference, the table also includes the performance of the original DT-type methods without augmentation, displayed in gray font. Performance changes due to augmentation are indicated with red upward arrows (↑) for improvements and green downward arrows (↓) for degradations compared to the original DT-type methods. All reported values are averaged over five random seeds (0, 1012, 2024, 3036, 4048).

## 4.3 Return Augmentation Methods for Off-Dynamics RL

Here we discuss how to implement REAG$_{MV}^*$ and REAG$_{Dara}^*$ in practice. We implement REAG$_{Dara}^*$ based on the dynamics-aware reward augmentation method proposed in Liu et al. (2022). For REAG$_{MV}^*$, it involves training the CQL model across both the Target and Source Environments to derive the respective value functions, denoted as $Q_T$ and $Q_S$. The derived value functions are then used to relabel the returns of trajectories in the original dataset. More specifically, the relabeled return $\hat{g}^S$ is calculated as defined in (3.4). Within this framework, we use $\mu^S(s, a)$ to denote $Q_S(s, a)$, and $Q_T(s, a)$ corresponds to $\mu^T(s, a)$. For the computation of $\sigma_S(s, a)$ and $\sigma_T(s, a)$, we employ the following methodology: For a given state $s$, we use the policy of CQL on the source dataset to obtain $n$ available actions $\{a_1^S, a_2^S, \ldots, a_n^S\}$ given the state $s$, with the corresponding $Q$ values $\{Q_S(s, a_1^S), Q_S(s, a_2^S), \ldots, Q_S(s, a_n^S)\}$, and $n$ available actions $\{a_1^T, a_2^T, \ldots, a_n^T\}$ in the target environment obtained from the CQL policy trained over the target dataset, with the corresponding $Q$ values $\{Q_T(s, a_1^T), Q_T(s, a_2^T), \ldots, Q_T(s, a_{n'}^T)\}$. The standard deviations $\sigma_S(s, a)$ and $\sigma_T(s, a)$ are then calculated as specified as follows.

$$\sigma_S(s, a) = \text{std}(Q_S(s, a_1^S), Q_S(s, a_2^S), \ldots, Q_S(s, a_n^S)),$$
$$\sigma_T(s, a) = \text{std}(Q_T(s, a_1^T), Q_T(s, a_2^T), \ldots, Q_T(s, a_n^T)).$$

For a detailed discussion, please refer to Section 3. As defined in (3.4), computing the ratio $\frac{\sigma_T(s,a)}{\sigma_S(s,a)}$ is essential. However, extreme values of this ratio can lead to instability during training. To address this, we introduce a clipping technique that constrains the ratio within an upper bound $\theta_1$ and a lower bound $\theta_2$. This helps stabilize REAG$_{MV}^*$ training by mitigating two key challenges. First, since this ratio depends on the variance of return-to-go in both the source and target environments, extreme variance values can introduce large gradients or noisy updates, destabilizing training. Second, variance is estimated using the Q-value function learned through CQL on the source and target datasets, which may introduce estimation errors. By bounding the ratio within a controlled range, clipping reduces the impact of these errors and prevents instability.

Table 2 presents a performance comparison of the REAG$_{MV}^*$ and REAG$_{Dara}^*$ return augmentation techniques integrated into different DT-type frameworks, including DT, Reinformer, and QT, in off-dynamics scenarios. The results demonstrate that both REAG$_{MV}^*$ and REAG$_{Dara}^*$ effectively enhance DT-type frameworks, improving performance in most off-dynamics scenarios compared to their original, non-augmented counterparts. Specifically, REAG$_{MV}^*$, which augments based on return values, leverages information from both the source and target environments, making it particularly

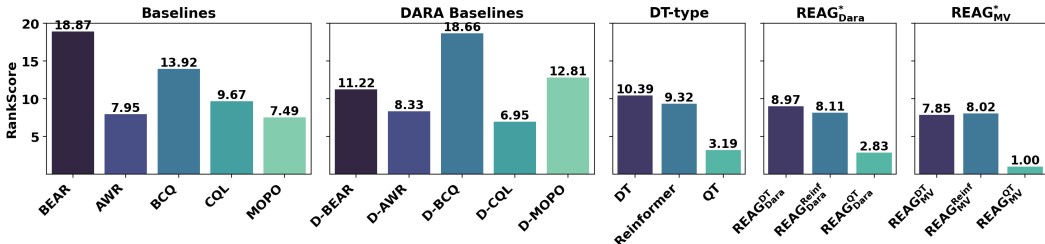

**Figure 1** Average normalized rank scores for all baseline algorithms across the Medium, Medium-Replay, and Medium-Expert datasets under BodyMass and JointNoise shift settings in the Walker2D, Hopper, and HalfCheetah environments. Within each setting, algorithms were ranked based on performance, with the top-performing algorithm assigned a rank of 1. Tied scores received the same rank, with subsequent ranks adjusted accordingly. Lower rank scores indicate better overall performance. The original ranks (from 19 algorithms) were normalized to a scale of 1 to 19. The figure presents the average normalized rank scores across the Walker2D, Hopper, and HalfCheetah environments.

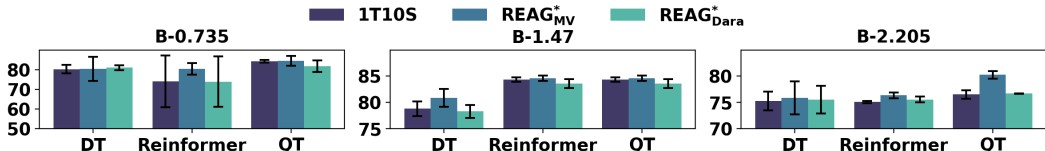

**a** Performance evaluation under varying body mass shift settings in the Walker2D medium environment.

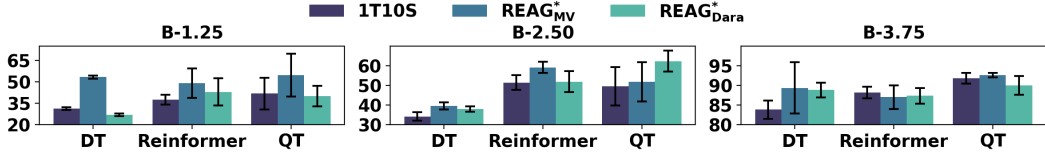

**b** Performance evaluation under varying body mass shift settings in the Hopper medium environment.

**Figure 2** Performance of $\mathbf{REAG^*_{MV}}$ and $\mathbf{REAG^*_{Dara}}$ algorithms under different body mass shift settings in the Walker2D and Hopper medium environments. **"B-x"** denotes that the body mass in the simulator is set to **x**. The target body mass is **2.94** in the Walker2D environment and **5** in the Hopper environment.

well-suited for return-based algorithms. In contrast, $REAG^*_{Dara}$, which augments based on reward values, exhibits more variable performance across different environments and dataset settings. While $REAG^*_{Dara}$ improves performance in certain cases, $REAG^*_{MV}$ consistently delivers more stable and robust improvements.

DARA is a widely adopted approach for addressing off-dynamics RL problems by introducing reward augmentation to enhance policy adaptation from a source dataset to a target environment while minimizing reliance on extensive target data. It seamlessly integrates with traditional offline RL frameworks such as CQL and BCQ. In our evaluation, we compare our proposed methods against DARA-based approaches, including both traditional RL frameworks and their DARA-augmented variants, as well as DT-type frameworks with and without $REAG^*_{MV}$ and $REAG^*_{Dara}$ augmentation, providing a comprehensive assessment of augmentation techniques for off-dynamics adaptation. We present a comparative ranking where lower average rank scores indicate better overall performance, as shown in Figure 1; for the raw results of each setting, please refer to Appendix E. The results demonstrate that DT-type frameworks exhibit strong potential in solving off-dynamics RL problems, outperforming traditional offline RL methods, particularly in the case of QT. Return-based augmentation techniques further enhance effectiveness, with $REAG^*_{MV}$ and $REAG^{QT}_{MV}$ achieving state-of-the-art performance compared to other baselines. Additionally, while DARA effectively improves the performance of non-return-based offline RL methods, a noticeable gap remains between these approaches and DT-type methods.

### 4.4 Ablation Studies for Return Augmentation Methods

In this section, we present an ablation study to examine the key factors influencing the performance of $REAG^*_{Dara}$ and $REAG^*_{MV}$. We focus on two aspects—**Dynamics Shift** and **Clipped Augmented**

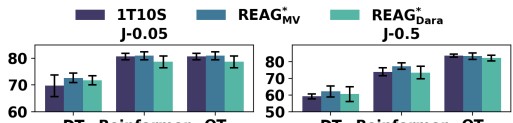
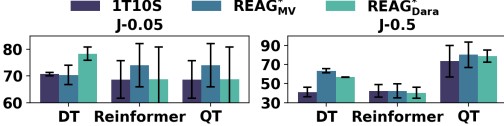

**a** Performance evaluation under varying joint noise shift settings in the Walker2D medium environment.

**b** Performance evaluation under varying joint noise shift settings in the Hopper medium environment.

**Figure 3** Performance of $\mathbf{REAG^*_{MV}}$ and $\mathbf{REAG^*_{Dara}}$ algorithms across varying JointNoise shift settings in the Walker2D and Hopper medium environments. **"J-x"** denotes the addition of random noise in the range **(-x, +x)** to the action.

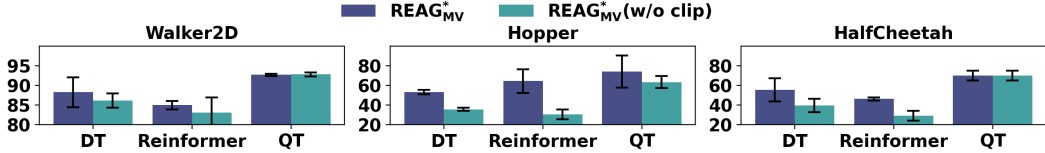

**Figure 4** Comparison of $REAG^*_{MV}$ with and without the clipping technique in the Medium Expert setting of the Walker2D environment under BodyMass shift. Results are averaged over five seeds.

**Return**—while the analysis of **Consistent Augmented Return** and **Return Learning** is deferred to Appendix F due to space limitations.

**Dynamics Shift.** To evaluate the impact of shifting source environments on $REAG^*_{MV}$ and $REAG^*_{Dara}$, we assess their performance under various BodyMass and JointNoise shift settings. The experimental results are presented in Figure 2 and Figure 3. Our findings indicate that as the body mass shift increases—creating a greater discrepancy from the target environment—performance deteriorates in both the Walker2D and Hopper Medium environments. Similarly, introducing higher levels of action noise leads to a decline in performance, suggesting that increased random noise raises the likelihood of failure, ultimately resulting in poorer outcomes. This performance degradation is particularly evident in the DT framework, highlighting its sensitivity to off-dynamics shifts, whereas Reinformer and QT demonstrate greater robustness. Across all shift experiments, $REAG^*_{MV}$ consistently outperforms $REAG^*_{Dara}$, with the performance gap becoming especially pronounced under larger shifts, such as in the Hopper environment with a body mass shift of 1.25.

**Clipped Augmented Return.** For data augmentation in $REAG^*_{MV}$, we apply a clipping technique to prevent the occurrence of extreme values. To evaluate its impact, we compare the performance of $REAG^*_{MV}$ with and without clipping in the Walker2D, Hopper, and HalfCheetah environments under BodyMass shifts with Medium Expert dataset. The results, presented in Figure 4, demonstrate that mitigating extreme values generally enhances the performance of $REAG^*_{MV}$. Additionally, we observe that for $REAG^{QT}_{MV}$, clipping does not yield significant improvements compared to DT and Reinformer. We hypothesize that this is due to the QT mechanism, which inherently regularizes the return, whereas DT and Reinformer lack such a mechanism.

## 5 Conclusion and Future Work

In this work, we introduced the Return-Augmented (REAG) method to improve Decision Transformer–type approaches in off-dynamics reinforcement learning by aligning source-domain returns with the target environment. We developed two practical variants, $REAG^*_{Dara}$ and $REAG^*_{MV}$, and provided theoretical guarantees showing that REAG trained on source data can achieve the same suboptimality as policies trained directly on target data. Empirical results confirm that REAG enhances DT-type baselines and outperforms several dynamic programming–based methods. Overall, REAG offers a promising direction for leveraging source-domain data to address challenges in offline, off-policy, and off-dynamics RL. Future work may extend REAG to more diverse environments and further refine its augmentation strategies.

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

## A  Related Work

**Off-dynamics reinforcement learning (RL).**  It is a type of domain adaptation problem in RL, drawing on concepts from transfer learning (Pan and Yang, 2009). There are many algorithms proposed to solve this problem (Niu et al., 2022; Liu et al., 2024; Xu et al., 2024). One of the promising approaches is to modify the reward in the source domain. The DARC algorithm (Eysenbach et al., 2020) addresses this domain adaptation challenge in the online setting by proposing a reward augmentation method that matches the optimal trajectory distribution between the source and target domains. Building on this, DARA (Liu et al., 2022) utilizes reward augmentation to supplement a limited target dataset with a larger source dataset. Unlike DARC and DARA, which are based on dynamic programming, our work adopts the adaptation setting of DARA and introduces a novel augmentation method tailored for RCSL, specifically focusing on the Decision Transformer. PAR (Lyu et al., 2024a) learns state encoder and state-action encoder utilizing the dynamics representation deviation to augment the reward in online settings.

**Return Conditioned Supervised Learning (RCSL).**  It is a general framework for powerful supervised methods in offline RL (Brandfonbrener et al., 2022). Notable works such as RvS (Emmons et al., 2021) and Decision Transformer (DT) (Chen et al., 2021) have shown competitive performance compared to traditional RL methods. The core idea of RCSL is to condition policies on a desired return. In this paper, we primarily focus on DT, which is a specific instance of RCSL and conducts offline RL through sequence generation. The generalization potential of DT has inspired researchers to explore its use in various settings. For example, Zheng et al. (2022); Xu et al. (2022) leverage the DT in the offline-to-online RL and meta RL respectively. However, no prior work has explicitly explored the adaptation capabilities of DT in the off-dynamics RL setting.

**Additional Related Work.**  Niu et al. (2022); Xu et al. (2024); Gui et al. (2023); Lyu et al., 2024b present recent advancements in off-dynamics RL methods. Specifically, H2O (Niu et al., 2022) performs importance weighting and penalizes Q-values with large dynamics gaps in offline-to-online settings. VGDF (Xu et al., 2024) filters data based on value consistency in online off-dynamics RL scenarios, while CPD (Gui et al., 2023) employs a dynamics alignment module to minimize discrepancies. PAR (Lyu et al.) addresses the off-dynamics problem by capturing representation mismatches. Lyu et al. (2024b) introduces a newly proposed off-dynamics RL benchmark, demonstrating that IQL achieves strong performance in off-dynamics RL settings. For cross-domain offline RL methods, BOSA (Liu et al., 2024) tackles OOD state actions with policy optimization and OOD dynamics with value optimization, IGDF (Wen et al., 2024) selectively shares transitions from the source domain via contrastive learning, and SRPO (Xue et al., 2024) learns the stationary state distribution to regularize the policy in a new environment.

## B  Sample Complexity of Off-Dynamics RCSL

In this section, we provide the rigorous analysis of the sample complexity of the off-dynamics RCSL. To this end, we first define some useful notations. We assume there are $N^S$ trajectories in the source dataset $\mathcal{D}^S$, and $N^T$ trajectories in the target dataset $\mathcal{D}^T$. Denote $P_\beta^S$ as the joint distribution of state, action, reward and return-to-go induced by the behavior policy $\beta$ in the source environment, and $P_\beta^T$ in the target environment. Denote $d_\pi^S$ as the marginal distribution of state $s$ induced by any policy $\pi$ in the source environment, and $d_\pi^T$ in the target environment.

Denote $J^T(\pi)$ as the expected cumulative reward under any policy $\pi$ and the target environment. For any return-to-go $g$ in the source dataset $\mathcal{D}^S$, we transform $g$ by an oracle defined in (3.3) with others remain the same, then we get a modified dataset $\tilde{D}^S$. We denote the mixed dataset as $\mathcal{D} = \mathcal{D}^T \cup \tilde{\mathcal{D}}^S$.

We first show the sample complexity of DT with only the samples from the target dataset $\mathcal{D}^T$. If we only use the offline dataset $\mathcal{D}^T$ collect from the target environment, i.e., at training time we minimizes the empirical negative log-likelihood loss:

$$\hat{L}^T(\pi) = - \sum_{\tau \in \mathcal{D}^T} \sum_{1 \le t \le H} \log \pi(a_t | s_t, g(s_t)).$$

Then we get the following sample complexity guarantee based on the result in Brandfonbrener et al. (2022).

**Corollary B.1.** There exists a conditioning function $f : \mathcal{S} \to \mathbb{R}$ such that assumptions (1)-(3) in Assumption 3.1, (1) and (2) in Assumption 3.2 hold. Further assume assumptions (1)-(3) in Assumption 3.3 hold. Then for some $\delta \in (0, 1)$, with probability at least $1 - \delta$, we have

$$J^T(\pi^\star) - J^T(\hat{\pi}_f) \leq O\Big(\frac{C_f}{\alpha_f} H^2 \Big(\sqrt{c}\Big(\frac{\log|\Pi|/\delta}{N^T}\Big)^{1/4} + \sqrt{\epsilon_{\text{approx}}}\Big) + \frac{\epsilon}{\alpha_f} H^2\Big).$$

Now we consider the case of mixed dataset, where we train our policy on both the target dataset and the source dataset using the proposed returned conditioned decision transformer methods. Note that the size of the target environment dataset is usually small, while the size of the source environment dataset is much larger, that is, $N^T \ll N^S$. If we incorporate the modified source dataset into the supervised learning, that is, we minimize the following empirical negative log-likelihood loss:

$$\hat{L}^{\text{mix}}(\pi) = -\sum_{\tau \in \mathcal{D}} \sum_{1 \leq t \leq H} \log \pi(a_t | s_t, g(s_t)). \tag{B.1}$$

An observation is that, with the modified source dataset, the regret $J^T(\pi^\star) - J^T(\hat{\pi}_f)$ can be significantly reduced. We state this observation in the following theorem, which is the formal version of Theorem 3.4.

**Theorem B.2.** There exists a conditioning function $f$ such that Assumptions 3.1 and 3.2 hold. Further assume Assumption 3.3 holds. Then for some $\delta \in (0, 1)$, with probability at least $1 - \delta$, we have

$$J^T(\pi^\star) - J^T(\hat{\pi}_f) \leq O\Big(\frac{C_f}{\alpha_f} \frac{N^S + N^T}{N^S/\gamma_f + N^T} H^2 \Big(\sqrt{c}\Big(\frac{\log|\Pi|/\delta}{N^T + N^S}\Big)^{1/4} + \sqrt{\epsilon_{\text{approx}}}\Big) + \frac{\epsilon}{\alpha_f} H^2\Big).$$
$$\tag{B.2}$$

**Remark B.3.** Compared to Corollary B.1, Theorem B.2 suggests that the modified samples from the source domain could enhance the performance of RCSL when the domain occupancy overlap coefficient $\gamma_f$ is large. In particular, when $N^S \gg N^T$ and $\gamma_f = O(1)$, (B.2) can be simplified to

$$J^T(\pi^\star) - J^T(\hat{\pi}_f) \leq O\Big(\frac{C_f}{\alpha_f} H^2 \Big(\sqrt{c}\Big(\frac{\log|\Pi|/\delta}{N^S}\Big)^{1/4} + \sqrt{\epsilon_{\text{approx}}}\Big) + \frac{\epsilon}{\alpha_f} H^2\Big),$$

which significantly improves the bound on suboptimality in Corollary B.1.

# C  Proof of Theorem B.2

**Lemma C.1** (Corollary 1 of Brandfonbrener et al. (2022)). Under the assumptions in Assumption 3.1, there exists a conditioning function such that

$$J^T(\pi^\star) - J^T(\pi_f^{\text{RCSL}}) \leq \epsilon\Big(\frac{1}{\alpha_f} + 3\Big)H^2.$$

**Lemma C.2** (Lemma 1 of Brandfonbrener et al. (2022)). For any two policies $\pi$, $\pi'$, we have

$$\big\|d_\pi^T - d_{\pi'}^T\big\|_1 \leq 2H \cdot \mathbb{E}_{s \sim d_\pi^T}\big[TV(\pi(\cdot|s)\|\hat{\pi}(\cdot|s))\big].$$

We define $d_\beta^{mix} = \frac{N^T}{N^T + N^S} d_\beta^T + \frac{N^S}{N^T + N^S} d_\beta^S$. Define

$$L(\hat{\pi}) = \mathbb{E}_{s \sim d_\beta^{mix}, g \sim P_\beta^T(\cdot|s)}\big[D_{\text{KL}}\big(P_\beta^T(\cdot|s,g)\|\hat{\pi}(\cdot|s,g)\big)\big].$$

**Theorem C.3.** Consider any function $f : \mathcal{S} \to \mathbb{R}$ such that the assumptions in Assumption 3.2 hold. Then for any estimated RCSL policy $\hat{\pi}$ that conditions on $f$ at test time (denoted by $\hat{\pi}_f$), we have

$$J^T(\pi_f^{RCSL}) - J^T(\hat{\pi}_f) \leq \frac{C_f \gamma_f}{\alpha_f} H^2 \sqrt{2L(\hat{\pi})}.$$

*Proof.* By definition and Lemma C.2, we have

$$J^T(\pi_f) - J^T(\hat{\pi}_f) = H\big(\mathbb{E}_{P_{\pi_f}^T}[r(s,a)] - \mathbb{E}_{P_{\hat{\pi}_f}^T}[r(s,a)]\big)$$
$$\leq H \cdot \|d_{\pi_f} - d_{\hat{\pi}_f}\|_1$$
$$\leq 2 \cdot \mathbb{E}_{s \sim d_{\pi_f}^T}\big[TV(\pi_f(\cdot|s)\|\hat{\pi}_f(\cdot|s))\big]H^2.$$

Next, we have

$$2 \cdot \mathbb{E}_{s \sim d_{\pi_f}^T}\big[TV(\pi_f(\cdot|s)||\hat{\pi}_f(\cdot|s))\big]$$

$$= \mathbb{E}_{s \sim d_{\pi_f}^T}\Big[\int_a \big|P_\beta^T(a|s, f(s)) - \hat{\pi}(a|s, f(s))|\big|\Big]$$

$$= \mathbb{E}_{s \sim d_{\pi_f}^T}\Big[\frac{P_\beta^T(f(s)|s)}{P_\beta^T(f(s)|s)} \int_a \big|P_\beta^T(a|s, f(s)) - \hat{\pi}(a|s, f(s))|\big|\Big]$$

$$\leq 2\frac{C_f}{\alpha_f}\mathbb{E}_{s \sim d_\beta^T, g \sim P_\beta^T(\cdot|s)}\big[TV(P_\beta^T(a|s, f(s))||\hat{\pi}(a|s, f(s)))\big]$$

$$\leq 2\frac{C_f}{\alpha_f}\frac{N^S + N^T}{N^S/\gamma_f + N^T} \cdot \mathbb{E}_{s \sim d_\beta^{mix}, g \sim P_\beta^T(\cdot|s)}\big[TV(P_\beta^T(a|s, f(s))||\hat{\pi}(a|s, f(s)))\big]$$

$$\leq \frac{C_f}{\alpha_f}\frac{N^S + N^T}{N^S/\gamma_f + N^T} \cdot \mathbb{E}_{s \sim d_\beta^{mix}, g \sim P_\beta^T(\cdot|s)}\Big[\sqrt{2KL(P_\beta^T(a|s, f(s))||\hat{\pi}(a|s, f(s)))}\Big]$$

$$\leq \frac{C_f}{\alpha_f}\frac{N^S + N^T}{N^S/\gamma_f + N^T}\sqrt{2L(\hat{\pi})}.$$

$\square$

*Proof of Theorem B.2.* Following the same argument in the proof of Corollary 3 in Brandfonbrener et al. (2022), we have

$$J^T(\pi_f^{\text{RCSL}}) - J^T(\hat{\pi}_f) \leq O\Big(2\frac{C_f}{\alpha_f}\frac{N^S + N^T}{N^S/\gamma_f + N^T}H^2\Big(\sqrt{c}\Big(\frac{\log|\Pi|/\delta}{N^S + N^T}\Big)^{1/4} + \sqrt{\epsilon_{\text{approx}}}\Big)\Big).$$

Invoking Lemma C.1, we have

$$J^T(\pi^\star) - J^T(\hat{\pi}_f) \leq O\Big(2\frac{C_f}{\alpha_f}\frac{N^S + N^T}{N^S/\gamma_f + N^T}H^2\Big(\sqrt{c}\Big(\frac{\log|\Pi|/\delta}{N^T + N^S}\Big)^{1/4} + \sqrt{\epsilon_{\text{approx}}}\Big) + \frac{\epsilon}{\alpha_f}H^2\Big).$$

This completes the proof. $\square$

# D  Detailed Experiment Setting

## D.1  Environment and Dataset

In this section, we provide details of the environments and datasets used in our experiments. We evaluate our approaches in the Hopper, Walker2D, and HalfCheetah environments, using the corresponding environments from Gym as our target environments.

### D.1.1  Target Environment Dataset Creation

For the target datasets, we construct two distinct datasets: one containing a smaller amount of data (1T) and another with a larger amount (10T). The 10T dataset consists of ten times the number of trajectories as the 1T dataset.

Both Liu et al. (2022) and our work aim to demonstrate the following two key points:

- The 10T dataset represents high-quality data, whereas the 1T dataset represents lower-quality data due to its smaller size.
- Off-dynamics RL algorithms can enhance performance on 1T by effectively leveraging 10S source domain data through appropriate data augmentation.

Liu et al. (2022) creates the 1T dataset by splitting the original target dataset (10T) based on timesteps, selecting the last 1/10 timesteps as 1T. However, this approach introduces unintended bias in the Medium Replay setting, where offline trajectories are collected from a replay buffer in which the behavior policy improves over time. Consequently, the final 1/10 timesteps tend to exhibit a higher average return than the overall 10T dataset, undermining the intended quality distinction between 1T and 10T.

To address this issue and ensure a fair evaluation of off-dynamics RL algorithms, we propose a uniform sampling method across trajectories in the target dataset. This approach ensures that the sampled 1T dataset is a representative subset of the target data, free from biases introduced by timestep-based selection. Notably, our method produces a 1T dataset of lower quality than that of Liu et al. (2022) in medium replay setting. If an off-dynamics RL algorithm can significantly improve performance on our 1T dataset and achieve results comparable to the original 10T dataset, it would serve as a more rigorous evaluation and a stronger indicator of the algorithm's effectiveness.

### D.1.2 Source Environment Dataset Creation

We employ BodyMass shift, JointNoise shift to construct the source environments. The following descriptions provide detailed insights into the process of creating these source environments.

- **BodyMass Shift:** The body mass of the agents is modified by adjusting the mass parameters in the Gym environment. Detailed body mass settings are provided in Table 3.

- **JointNoise Shift:** Noise is introduced to the agents' joints by adding perturbations to the actions during source data collection. Specifically, the noise is sampled uniformly from the range $[-0.05, +0.05]$ and applied to the actions when generating the source offline dataset. Detailed joint noise settings are provided in Table 3.

For the source datasets, we utilize the BodyMass Shift and JointNoise Shift datasets from (Liu et al., 2022). Additionally, in our ablation study, we explore variations of BodyMass and JointNoise shifts beyond those specified in Table 3. We also collect medium-level source datasets for the Hopper, Walker2D, and HalfCheetah environments. Behavior policies are generated by training agents with SAC using rlkit (https://github.com/vitchyr/rlkit), with checkpoints used for dataset collection. We construct the Random, Medium, Medium Replay, and Medium Expert datasets, each reflecting different performance levels determined by their corresponding SAC checkpoints. For the JointNoise Shift setting, instead of training a new SAC policy and collecting data through environment interaction, we introduce random noise within a specified range directly to the actions.

**Table 3** BodyMass Shift and JointNoise Shift in Hopper, Walker2D and HalfCheetah.

|  | Hopper | | Walker2D | | HalfCheetah | |
|---|---|---|---|---|---|---|
|  | **BodyMass** | **JointNoise** | **BodyMass** | **JointNoise** | **BodyMass** | **JointNoise** |
| **Source** | mass[-1]=2.5 | action[-1]+noise | mass[-1]=1.47 | action[-1]+noise | mass[4]=0.5 | action[-1]+noise |
| **Target** | mass[-1]=5.0 | action[-1]+0 | mass[-1]=2.94 | action[-1]+0 | mass[4]=1.0 | action[-1]+0 |

### D.2 Baselines

In our experiments, we use BEAR (Kumar et al., 2019), AWR (Peng et al., 2019), BCQ (Fujimoto et al., 2019), CQL (Kumar et al., 2020), and MOPO (Yu et al., 2020), along with their DARA-augmented variants (Liu et al., 2022), as baseline methods. We compare these baselines against DT (Chen et al., 2021), Reinformer (Zhuang et al., 2024), and QT (Hu et al., 2024), as well as our proposed REAG approaches.

### D.3 Hyperparameters

In this section, we outline the hyperparameters used for our REAG methodologies. The REAG approaches begin with dataset augmentation using either the DARA algorithm ($\text{REAG}^*_{\text{Dara}}$) or the Direct Matching of Return Distributions technique ($\text{REAG}^*_{\text{MV}}$). The augmented dataset is then used to train the DT-type frameworks, which is subsequently evaluated in the target environment. Specifically, for $\text{REAG}^*_{\text{Dara}}$, dataset augmentation follows the DARA algorithm, with its corresponding hyperparameters provided in Table 4. For $\text{REAG}^*_{\text{MV}}$, the augmentation process is described in Section 4.3, where a well-trained Conservative Q-Learning (CQL) model estimates state values, incorporating a clipping mechanism to mitigate extreme values. The hyperparameters for CQL training are provided in Table 5, the clipping ratios are listed in Table 6, and the training parameters for DT, Reinformer, and QT adhere to the settings from their respective original papers.

**Table 4** Hyperparameters used in the DARA algorithm.

| Hyperparameter | Value |
|---|---|
| **SA Discriminator MLP Layers** | 4 |
| **SAS Discriminator MLP Layers** | 4 |
| **Hidden Dimension** | 256 |
| **Nonlinearity Function** | ReLU |
| **Optimizer** | RMSprop |
| **Batch Size** | 256 |
| **Learning Rate** | $3 \times 10^{-4}$ |
| **$\Delta r$ Coefficient $\eta$** | 0.1 |

**Table 5** Hyperparameters used in the CQL algorithm.

| Hyperparameter | Value |
|---|---|
| **Actor MLP Layers** | 3 |
| **Critic MLP Layers** | 3 |
| **Hidden Dimension** | 256 |
| **Nonlinearity Function** | ReLU |
| **Optimizer** | Adam |
| **Batch size** | 256 |
| **Discount Factor** | 0.99 |
| **Temperature** | 1.0 |
| **Actor Learning rate** | $1 \times 10^{-4}$ |
| **Critic Learning rate** | $3 \times 10^{-4}$ |

**Table 6** Hyperparameters for the Clipping Technique Employed in the $\text{REAG}^*_{\text{MV}}$ Algorithm.

| Dataset | Clipping Ratio |
|---|---|
| **Walker2D Random** | $0.9 < \theta < 1.25$ |
| **Walker2D Medium** | $0.9 < \theta < 1.25$ |
| **Walker2D Medium Replay** | $0.9 < \theta < 1.25$ |
| **Walker2D Medium Expert** | $0.9 < \theta < 1.25$ |
| **Hopper Random** | $0.9 < \theta < 1$ |
| **Hopper Medium** | $0.9 < \theta < 1$ |
| **Hopper Medium Replay** | $0.9 < \theta < 1$ |
| **Hopper Medium Expert** | $0.9 < \theta < 1$ |
| **HalfCheetah Random** | $0.67 < \theta < 1.5$ |
| **HalfCheetah Medium** | $0.67 < \theta < 1.5$ |
| **HalfCheetah Medium Replay** | $0.67 < \theta < 1.5$ |
| **HalfCheetah Medium Expert** | $0.67 < \theta < 1.5$ |

| | | BEAR | | | AWR | | | BCQ | | | CQL | | |
|---|---|---|---|---|---|---|---|---|---|---|---|---|---|
| | | M | M-R | M-E | M | M-R | M-E | M | M-R | M-E | M | M-R | M-E |
| **W2D** | 1T | $4.638_{\pm3.882}$ | $0.777_{\pm0.105}$ | $9.267_{\pm1.692}$ | $68.023_{\pm1.687}$ | $28.426_{\pm2.974}$ | $100.566_{\pm0.513}$ | $62.567_{\pm2.459}$ | $60.638_{\pm0.683}$ | $101.610_{\pm1.309}$ | $65.618_{\pm2.818}$ | $57.402_{\pm6.161}$ | $101.611_{\pm0.143}$ |
| | 10T | $13.143_{\pm3.016}$ | $5.852_{\pm0.168}$ | $21.383_{\pm1.237}$ | $78.060_{\pm0.772}$ | $58.286_{\pm1.684}$ | $109.154_{\pm0.976}$ | $74.735_{\pm1.184}$ | $64.735_{\pm2.555}$ | $101.840_{\pm1.962}$ | $78.191_{\pm1.839}$ | $80.145_{\pm2.286}$ | $101.840_{\pm0.467}$ |
| **Hp** | 1T | $8.770_{\pm0.402}$ | $5.264_{\pm0.283}$ | $31.968_{\pm1.213}$ | $55.269_{\pm2.254}$ | $54.259_{\pm1.295}$ | $54.098_{\pm1.165}$ | $63.308_{\pm0.418}$ | $68.448_{\pm0.251}$ | $62.287_{\pm1.689}$ | $74.489_{\pm1.061}$ | $71.401_{\pm2.106}$ | $82.071_{\pm0.483}$ |
| | 10T | $20.398_{\pm2.102}$ | $5.554_{\pm0.842}$ | $88.236_{\pm2.192}$ | $64.494_{\pm2.217}$ | $57.548_{\pm1.778}$ | $105.361_{\pm1.392}$ | $73.462_{\pm2.527}$ | $60.385_{\pm0.418}$ | $102.775_{\pm1.912}$ | $82.945_{\pm0.323}$ | $73.168_{\pm2.712}$ | $102.071_{\pm1.759}$ |
| **Hc** | 1T | $2.659_{\pm0.167}$ | $1.602_{\pm0.275}$ | $3.089_{\pm0.104}$ | $41.672_{\pm0.732}$ | $28.023_{\pm4.027}$ | $90.168_{\pm1.398}$ | $41.051_{\pm2.908}$ | $25.828_{\pm6.142}$ | $60.173_{\pm4.175}$ | $44.393_{\pm0.263}$ | $26.955_{\pm1.274}$ | $61.621_{\pm13.093}$ |
| | 10T | $10.657_{\pm0.271}$ | $19.588_{\pm0.453}$ | $16.160_{\pm0.208}$ | $42.209_{\pm0.611}$ | $41.041_{\pm0.729}$ | $90.212_{\pm2.259}$ | $46.188_{\pm0.423}$ | $38.575_{\pm2.060}$ | $95.535_{\pm4.042}$ | $49.382_{\pm0.338}$ | $46.966_{\pm0.372}$ | $87.683_{\pm7.753}$ |

| | | MOPO | | | DT | | | Reinformer | | | QT | | |
|---|---|---|---|---|---|---|---|---|---|---|---|---|---|
| | | M | M-R | M-E | M | M-R | M-E | M | M-R | M-E | M | M-R | M-E |
| **W2D** | 1T | $20.953_{\pm2.715}$ | $20.313_{\pm3.488}$ | $20.569_{\pm0.983}$ | $67.261_{\pm2.316}$ | $34.482_{\pm5.890}$ | $107.171_{\pm1.611}$ | $79.034_{\pm1.506}$ | $38.072_{\pm9.174}$ | $103.284_{\pm5.437}$ | $81.756_{\pm1.671}$ | $67.546_{\pm9.505}$ | $111.722_{\pm1.398}$ |
| | 10T | $22.261_{\pm2.811}$ | $18.529_{\pm1.760}$ | $21.196_{\pm3.103}$ | $79.697_{\pm3.348}$ | $68.528_{\pm1.924}$ | $108.622_{\pm1.815}$ | $81.377_{\pm1.903}$ | $68.168_{\pm2.661}$ | $109.845_{\pm0.726}$ | $88.262_{\pm12.886}$ | $85.092_{\pm8.727}$ | $111.394_{\pm0.469}$ |
| **Hp** | 1T | $31.038_{\pm2.868}$ | $5.849_{\pm0.146}$ | $35.099_{\pm1.212}$ | $66.073_{\pm1.745}$ | $61.686_{\pm2.592}$ | $100.719_{\pm1.679}$ | $74.737_{\pm4.807}$ | $36.008_{\pm6.575}$ | $60.753_{\pm14.433}$ | $70.927_{\pm6.482}$ | $83.406_{\pm4.734}$ | $108.225_{\pm5.596}$ |
| | 10T | $32.769_{\pm1.788}$ | $8.638_{\pm1.395}$ | $36.161_{\pm2.204}$ | $85.589_{\pm5.311}$ | $69.701_{\pm5.317}$ | $108.087_{\pm1.049}$ | $77.792_{\pm4.652}$ | $39.856_{\pm12.334}$ | $79.389_{\pm28.054}$ | $90.176_{\pm0.186}$ | $100.321_{\pm1.121}$ | $112.908_{\pm3.154}$ |
| **Hc** | 1T | $64.329_{\pm2.096}$ | $12.277_{\pm1.953}$ | $25.055_{\pm7.834}$ | $41.204_{\pm0.430}$ | $15.164_{\pm4.847}$ | $77.500_{\pm3.323}$ | $42.958_{\pm0.065}$ | $18.493_{\pm1.584}$ | $72.085_{\pm3.491}$ | $50.464_{\pm0.127}$ | $32.318_{\pm2.435}$ | $87.854_{\pm6.657}$ |
| | 10T | $65.863_{\pm1.289}$ | $59.724_{\pm1.056}$ | $28.221_{\pm6.078}$ | $42.273_{\pm0.379}$ | $34.508_{\pm1.482}$ | $82.844_{\pm7.635}$ | $43.243_{\pm0.262}$ | $39.434_{\pm0.362}$ | $87.378_{\pm3.340}$ | $51.284_{\pm0.605}$ | $49.587_{\pm0.334}$ | $94.116_{\pm0.321}$ |

**Table 7** Performance comparison of algorithms on the **1T** and **10T** datasets. The experiments are conducted in the **Walker2D (W2D)**, **Hopper (Hp)**, and **HalfCheetah (Hc)** using the **Medium (M)**, **Medium Replay (M-R)**, and **Medium Expert (M-E)** datasets. All reported values are averaged over five seeds (0, 1012, 2024, 3036, 4048).

# E    Additional Experiments Results

This section presents more comprehensive experimental results, including additional variance information.

In Table 1, we present the partial performance of various algorithms and their DARA variants in the Walker2D medium environment under BodyMass and JointNoise shift settings, considering both limited and sufficient target data scenarios. The complete experimental results are provided in Table 7. Additionally, Table 8 and Table 9 present a comprehensive comparison of different algorithms and their corresponding augmented variants in addressing the off-dynamics problem across various environments and shift settings.

# F    Ablation Study

**Consistent Augmented Return.**    It is worth noting that our augmented target returns do not satisfy the *consistency condition*, which requires that the augmented returns follow $R_{t+1} - R_t = r_t$, as enforced by the original DT. To verify whether consistency is a necessary condition for augmentation in off-dynamics settings, we conduct the following ablation study. Specifically, we introduce a variant of $\text{REAG}^*_{\text{MV}}$, denoted as $\text{REAG}^*_{\text{MV}}$ (consistent), where for each trajectory in the target environment, return augmentation is applied only to the initial return, while all subsequent augmented returns are derived using the consistency condition $R_{t+1} - R_t = r_t$. The results, presented in Figure 5, indicate that $\text{REAG}^*_{\text{MV}}$ outperforms its consistency-enforced variant in most cases. This finding suggests that enforcing consistency does not necessarily improve performance; instead, it may limit the effectiveness of $\text{REAG}^*_{\text{MV}}$ in the context of off-dynamics offline reinforcement learning.

**Return Learning.**    To evaluate the learned value functions, $Q_S$ and $Q_T$, and their impact on $\text{REAG}^*_{\text{MV}}$, we conduct an ablation experiment. Specifically, we assess the quality of the learned

| | | BEAR | AWR | BCQ | CQL | MOPO | D-BEAR | D-AWR | D-BCQ | D-CQL | D-MOPO |
|---|---|---|---|---|---|---|---|---|---|---|---|
| Walker2D M | BM | 5.776 ± 1.653 | 77.442 ± 0.340 | 70.681 ± 0.539 | 73.317 ± 1.368 | 21.617 ± 1.277 | 6.516 ± 3.220 | 78.004 ± 0.911 | 72.023 ± 0.695 | 74.276 ± 2.582 | 21.621 ± 1.063 |
| | JN | 4.926 ± 1.418 | 67.636 ± 1.468 | 62.696 ± 1.037 | 68.962 ± 0.865 | 23.552 ± 1.063 | 6.933 ± 1.884 | 64.303 ± 0.513 | 60.681 ± 1.118 | 69.141 ± 0.944 | 23.57 ± 0.665 |
| Walker2D M-R | BM | 0.0668 ± 4.951 | 47.033 ± 2.278 | 50.714 ± 1.918 | 54.753 ± 0.335 | 11.563 ± 2.751 | 1.078 ± 2.083 | 32.008 ± 1.286 | 51.447 ± 3.108 | 57.432 ± 0.764 | 12.129 ± 2.755 |
| | JN | 0.474 ± 0.719 | 31.623 ± 2.551 | 50.601 ± 1.611 | 50.600 ± 1.589 | 11.379 ± 0.596 | 0.384 ± 3.823 | 36.807 ± 2.442 | 50.714 ± 0.876 | 51.742 ± 1.061 | 15.389 ± 0.559 |
| Walker2D M-E | BM | 19.799 ± 3.116 | 110.324 ± 1.053 | 112.343 ± 1.488 | 107.187 ± 3.209 | 18.324 ± 0.708 | 17.491 ± 2.844 | 109.743 ± 2.632 | 113.069 ± 1.602 | 105.401 ± 2.186 | 20.741 ± 0.399 |
| | JN | 14.225 ± 1.338 | 104.662 ± 2.370 | 112.926 ± 1.491 | 104.019 ± 0.294 | 17.429 ± 0.639 | 14.203 ± 1.602 | 108.915 ± 1.915 | 111.249 ± 1.092 | 108.236 ± 1.206 | 19.325 ± 3.119 |
| Hopper M | BM | 22.436 ± 0.103 | 25.843 ± 0.325 | 24.853 ± 1.615 | 49.094 ± 2.207 | 20.765 ± 3.350 | 25.608 ± 1.063 | 26.594 ± 1.267 | 26.487 ± 1.366 | 45.101 ± 0.342 | 21.495 ± 0.848 |
| | JN | 8.536 ± 1.965 | 57.021 ± 0.938 | 74.559 ± 0.605 | 71.495 ± 0.126 | 23.556 ± 1.327 | 10.576 ± 2.052 | 61.463 ± 0.702 | 74.853 ± 0.626 | 63.611 ± 1.136 | 24.992 ± 0.944 |
| Hopper M-R | BM | 6.282 ± 0.132 | 55.607 ± 2.310 | 64.519 ± 0.813 | 66.455 ± 0.636 | 5.504 ± 1.701 | 2.619 ± 0.128 | 44.883 ± 1.595 | 64.168 ± 0.291 | 68.163 ± 0.559 | 5.482 ± 1.061 |
| | JN | 1.841 ± 3.814 | 37.821 ± 1.205 | 65.103 ± 0.703 | 61.302 ± 1.207 | 5.498 ± 0.568 | 5.637 ± 0.291 | 63.937 ± 3.879 | 64.519 ± 1.102 | 63.178 ± 1.218 | 6.147 ± 0.157 |
| Hopper M-E | BM | 22.934 ± 3.022 | 57.595 ± 0.612 | 109.367 ± 0.834 | 70.467 ± 2.712 | 30.541 ± 3.616 | 31.090 ± 0.463 | 78.262 ± 0.239 | 110.014 ± 2.153 | 72.149 ± 1.934 | 30.540 ± 0.842 |
| | JN | 39.031 ± 1.079 | 74.708 ± 1.889 | 108.639 ± 2.028 | 72.512 ± 0.781 | 30.537 ± 0.842 | 33.052 ± 0.385 | 60.952 ± 0.879 | 111.587 ± 1.602 | 94.128 ± 1.213 | 32.589 ± 1.985 |
| HalfCheetah M | BM | 5.431 ± 1.518 | 42.293 ± 0.862 | 39.835 ± 0.427 | 37.081 ± 0.358 | 58.457 ± 1.449 | 6.009 ± 1.705 | 41.800 ± 0.830 | 39.333 ± 0.506 | 37.189 ± 0.218 | 59.311 ± 0.949 |
| | JN | 1.948 ± 1.058 | 41.992 ± 0.762 | 50.511 ± 0.371 | 49.046 ± 0.420 | 61.073 ± 0.315 | 2.901 ± 0.402 | 42.545 ± 0.731 | 52.149 ± 0.457 | 49.284 ± 0.570 | 61.447 ± 0.734 |
| HalfCheetah M-R | BM | 7.425 ± 1.307 | 15.988 ± 5.339 | 32.553 ± 1.258 | 37.508 ± 0.520 | 50.429 ± 1.306 | 4.909 ± 0.562 | 17.918 ± 3.701 | 32.095 ± 1.258 | 37.721 ± 0.440 | 52.609 ± 0.621 |
| | JN | 18.337 ± 0.498 | 31.742 ± 4.199 | 46.567 ± 2.563 | 51.566 ± 0.246 | 51.918 ± 1.584 | 17.929 ± 0.479 | 38.125 ± 1.775 | 49.066 ± 0.645 | 52.991 ± 0.438 | 51.258 ± 1.709 |
| HalfCheetah M-E | BM | 4.356 ± 0.431 | 88.155 ± 1.836 | 61.771 ± 4.610 | 61.104 ± 4.131 | 51.040 ± 4.461 | 2.948 ± 0.691 | 89.201 ± 2.419 | 63.465 ± 3.303 | 62.665 ± 5.326 | 56.616 ± 2.609 |
| | JN | 3.195 ± 0.391 | 88.647 ± 2.669 | 62.486 ± 10.025 | 84.090 ± 1.109 | 54.630 ± 10.104 | 8.789 ± 0.271 | 89.220 ± 1.800 | 71.007 ± 4.201 | 84.210 ± 0.506 | 60.014 ± 7.011 |

**Table 8** Performance comparison of traditional offline reinforcement learning algorithms, including BEAR, AWR, BCQ, CQL, and MOPO, along with their DARA-augmented variants, under BodyMass and JointNoise distribution shifts in the Walker2D, Hopper, and HalfCheetah environments. Evaluations are conducted using the Medium (M), Medium Replay (M-R), and Medium Expert (M-E) settings of the **1T10S** dataset. The **1T10S** dataset comprises a **1T** (target) dataset and a **10S** (source) dataset. **"D-XX"** denotes the DARA-augmented variant of the **'XX'** algorithm.

| | | DT | Reinformer | QT | $\text{REAG}_{\text{MV}}^{\text{DT}}$ | $\text{REAG}_{\text{MV}}^{\text{Reinf}}$ | $\text{REAG}_{\text{MV}}^{\text{QT}}$ | $\text{REAG}_{\text{Dara}}^{\text{DT}}$ | $\text{REAG}_{\text{Dara}}^{\text{Reinf}}$ | $\text{REAG}_{\text{Dara}}^{\text{QT}}$ |
|---|---|---|---|---|---|---|---|---|---|---|
| Walker2D M | BM | 78.768 ± 1.233 | 80.857 ± 0.509 | 84.325 ± 0.425 | 80.857 ± 1.715 | 82.354 ± 1.479 | 84.582 ± 0.507 | 78.257 ± 2.423 | 80.666 ± 0.505 | 83.068 ± 0.859 |
| | JN | 71.068 ± 1.022 | 74.748 ± 1.721 | 80.621 ± 1.143 | 75.008 ± 1.834 | 75.008 ± 0.986 | 80.904 ± 1.502 | 71.779 ± 1.706 | 74.268 ± 1.341 | 78.672 ± 2.201 |
| Walker2D M-R | BM | 73.664 ± 1.920 | 67.032 ± 5.767 | 87.292 ± 0.631 | 73.708 ± 1.570 | 50.296 ± 14.211 | 87.491 ± 1.226 | 67.565 ± 0.799 | 66.658 ± 4.303 | 76.169 ± 7.567 |
| | JN | 58.255 ± 3.181 | 54.801 ± 3.217 | 82.139 ± 1.029 | 55.722 ± 2.653 | 47.591 ± 10.244 | 82.363 ± 4.206 | 62.226 ± 0.383 | 55.438 ± 4.833 | 79.795 ± 4.708 |
| Walker2D M-E | BM | 84.430 ± 0.823 | 83.388 ± 0.806 | 93.082 ± 0.348 | 88.235 ± 1.886 | 84.897 ± 1.117 | 92.744 ± 0.499 | 85.328 ± 0.865 | 83.761 ± 0.735 | 94.578 ± 1.383 |
| | JN | 115.746 ± 1.116 | 117.360 ± 2.550 | 116.149 ± 1.640 | 111.060 ± 2.247 | 118.218 ± 1.460 | 118.564 ± 0.697 | 111.236 ± 0.914 | 117.765 ± 2.499 | 116.115 ± 1.889 |
| Hopper M | BM | 34.057 ± 0.177 | 51.357 ± 3.713 | 49.516 ± 9.798 | 39.435 ± 1.239 | 59.085 ± 2.791 | 51.796 ± 9.971 | 37.787 ± 1.914 | 51.771 ± 5.322 | 62.262 ± 5.348 |
| | JN | 70.685 ± 0.726 | 70.340 ± 4.633 | 68.656 ± 7.079 | 70.356 ± 3.657 | 72.346 ± 5.877 | 73.987 ± 8.080 | 78.325 ± 2.522 | 70.466 ± 3.728 | 68.709 ± 12.160 |
| Hopper M-R | BM | 64.216 ± 1.504 | 17.534 ± 6.725 | 69.460 ± 13.948 | 66.092 ± 0.233 | 20.952 ± 9.794 | 76.287 ± 7.810 | 60.393 ± 1.086 | 27.238 ± 12.735 | 82.786 ± 11.992 |
| | JN | 61.870 ± 0.249 | 41.820 ± 15.773 | 93.704 ± 7.559 | 77.825 ± 1.638 | 43.985 ± 5.075 | 93.409 ± 4.696 | 83.525 ± 1.728 | 52.052 ± 10.035 | 51.456 ± 12.168 |
| Hopper M-E | BM | 33.554 ± 0.846 | 68.973 ± 7.512 | 61.162 ± 3.767 | 52.873 ± 0.454 | 64.206 ± 12.073 | 73.952 ± 16.294 | 33.631 ± 1.605 | 73.363 ± 7.674 | 77.279 ± 18.607 |
| | JN | 108.254 ± 1.583 | 109.256 ± 0.126 | 109.056 ± 0.214 | 109.367 ± 1.084 | 109.472 ± 0.103 | 109.803 ± 0.609 | 108.261 ± 2.612 | 109.255 ± 0.188 | 109.746 ± 0.771 |
| HalfCheetah M | BM | 39.954 ± 0.260 | 37.353 ± 0.483 | 44.656 ± 0.643 | 40.250 ± 0.911 | 42.451 ± 0.491 | 47.303 ± 0.318 | 37.599 ± 0.395 | 38.261 ± 1.238 | 46.383 ± 0.358 |
| | JN | 47.725 ± 0.431 | 48.274 ± 0.191 | 56.213 ± 0.327 | 44.149 ± 3.672 | 43.009 ± 0.307 | 52.394 ± 1.413 | 47.833 ± 0.284 | 48.404 ± 0.168 | 55.026 ± 0.410 |
| HalfCheetah M-R | BM | 20.966 ± 9.607 | 31.584 ± 1.248 | 41.300 ± 0.787 | 27.812 ± 3.256 | 32.114 ± 1.455 | 42.405 ± 0.729 | 24.059 ± 2.271 | 26.995 ± 4.373 | 41.359 ± 0.985 |
| | JN | 36.509 ± 4.414 | 40.296 ± 2.914 | 53.763 ± 0.793 | 38.417 ± 4.068 | 40.840 ± 2.880 | 53.870 ± 0.981 | 38.031 ± 3.529 | 38.436 ± 3.377 | 53.257 ± 0.586 |
| HalfCheetah M-E | BM | 54.981 ± 1.147 | 40.568 ± 0.984 | 71.008 ± 8.802 | 56.228 ± 2.930 | 46.048 ± 1.657 | 69.819 ± 5.120 | 51.357 ± 8.231 | 55.818 ± 1.849 | 76.533 ± 8.022 |
| | JN | 70.573 ± 8.599 | 76.073 ± 3.878 | 82.961 ± 4.019 | 77.762 ± 2.099 | 79.390 ± 0.149 | 83.692 ± 0.699 | 77.751 ± 2.702 | 78.981 ± 1.198 | 82.148 ± 2.758 |

**Table 9** Performance comparison of traditional offline reinforcement learning algorithms, including DT, Reinformer and QT, along with our proposed methods $\text{REAG}_{\text{MV}}^{\text{DT}}$, $\text{REAG}_{\text{Dara}}^{\text{DT}}$, $\text{REAG}_{\text{MV}}^{\text{Reinf}}$, $\text{REAG}_{\text{Dara}}^{\text{Reinf}}$, $\text{REAG}_{\text{MV}}^{\text{QT}}$ and $\text{REAG}_{\text{Dara}}^{\text{QT}}$ under BodyMass and JointNoise distribution shifts in the Walker2D, Hopper, and HalfCheetah environments. Evaluations are conducted using the Medium (M), Medium Replay (M-R), and Medium Expert (M-E) settings of the **1T10S** dataset. The **1T10S** dataset comprises a **1T** (target) dataset and a **10S** (source) dataset.

value functions in both the source and target domains. We select the Hopper environment with a medium-expert offline dataset as the target domain and the BodyMass shift as the source domain. Ideally, the value functions $Q_S$ and $Q_T$ learned through $\text{REAG}_{\text{MV}}^*$ should accurately reflect the returns of trajectories in their respective domains. To verify this, we train two additional DTs separately on the source and target offline datasets to obtain policies for these environments. Using these policies, we generate test trajectories through rollouts and then leverage the learned value functions $Q_S$ and $Q_T$, trained on the 10S and 1T datasets, to predict the returns of these test trajectories. By comparing the predicted returns with the actual returns, we assess the accuracy of the learned value functions. As shown in Figure 6, our learned value functions $Q_S$ and $Q_T$ accurately reflect the returns of trajectories

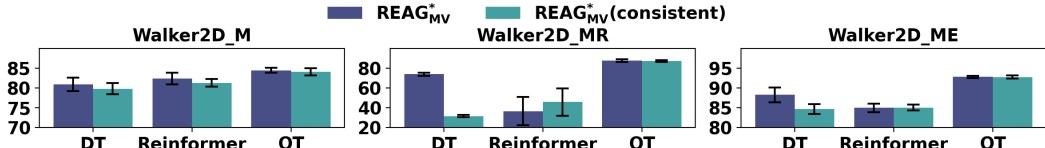

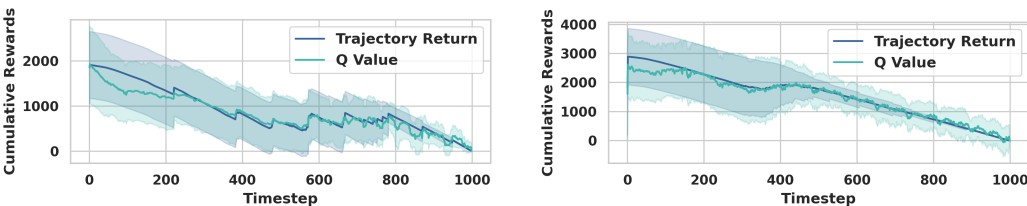

**Figure 5** Comparison of $REAG^*_{MV}$ and $REAG^*_{MV}$(consistent) across Medium, Medium Replay, and Medium Expert settings in the Walker2D environment under BodyMass shift. Results are averaged over five seeds.

**a** Comparison between cumulative rewards and estimated $Q_S$ values in the source environment with 100 trajectories.

**b** Comparison between cumulative rewards and estimated $Q_T$ values in the target environment with 100 trajectories.

**Figure 6** Comparison of the cumulative returns and the learned $Q$-values for the source (left) and target (right) environments using CQL. Results are plotted with the mean and variance of 100 trajectories.

605 collected by the policies in the source and target environments, demonstrating that the Q-values used
606 in our approach serve as reliable approximations.

