# OpenReview forum: "Return Augmented Decision Transformer for Off-Dynamics Reinforcement Learning"
_NeurIPS.cc/2025/Workshop/Reliable_ML — NeurIPS 2025 - Reliable ML Workshop_

### Official Review · Reviewer_3VRC · 2025-09-19
**Good Technical Contribution and Experiments but Need Clarity**

**Rating:** 7
**Confidence:** 3

**Review:**

This paper considers the offline off-dynamics RL problem, where the planner have access to two datasets $\mathcal D^S,\mathcal D^T$ with $\lvert \mathcal D^S\rvert\gg \lvert \mathcal D^T\rvert$. These two datasets, both generated by a behavior policy $\beta$, come from two MDPs $M^S,M^T$ sharing the same reward $r$ but different transitions $p^S,p^T$. The objective is to find a $\pi$ that behaves well on $M^T$ while utilizing those data from $M^S$.

This paper studies an important problem, presents a technically interesting and non-trivial new algorithm, a not-too-bad convergence theorem, and extensive experiments. I feel this paper is good enough to be presented at a workshop. But on the other hand, it's not super clear what can be a takeaway for the reliable ML community; I'd leave this question to the authors when preparing their poster / slides.

Furthermore, the introduction and preliminary parts are a bit too involved for general audience unfamiliar with the problem of off-dynamics RL to follow. I originally thought it's a 4-page short track submission, but it actually turns out to be the 9-page long track. I feel the authors could try devoting more space to a detailed introduction of DT and technical comparison with several related works in the revision.

---

### Official Review · Reviewer_whvb · 2025-09-20
**Good paper**

**Rating:** 7
**Confidence:** 4

**Review:**

Summary:
The paper proposes Return Augmented (REAG) method, which addresses the challenges of offline off-dynamics reinforcement learning (RL). This is a setting where a policy needs to be learned for a target environment with limited data, but a large amount of data is available from a different source environment with distinct dynamics.

Strengths:
1. The paper introduces a novel return augmentation method for Decision Transformer-type frameworks.
2. The authors provide a rigorous theoretical analysis. The formal theorem and its proof show that the policy learned with REAG can achieve comparable suboptimality to one learned on a large, clean target dataset.
3. Comprehensive experiments and ablation studies are conducted, showing that the proposed methods are effective.

Weakness:
1. $REAG_{MV}$ relies on training a CQL model to estimate Q-values and variances, introducing a dependency on another complex algorithm.
2. Strong assumptions, such as bounded occupancy mismatch and return coverage.

Suggestions:
1. Fonts in table 1 and 2 are too small, hard to read.